# Neutrophil Cell Shape Change: Mechanism and Signalling during Cell Spreading and Phagocytosis

**DOI:** 10.3390/ijms20061383

**Published:** 2019-03-19

**Authors:** Rhiannon E. Roberts, Maurice B. Hallett

**Affiliations:** Neutrophil Signalling Group, School of Medicine, Cardiff University, Cardiff CF14 4XN, UK

**Keywords:** neutrophils, phagocytosis, cell spreading, Ca^2+^, calpain, membrane expansion, membrane tension, ezrin, cortical actin

## Abstract

Perhaps the most important feature of neutrophils is their ability to rapidly change shape. In the bloodstream, the neutrophils circulate as almost spherical cells, with the ability to deform in order to pass along narrower capillaries. Upon receiving the signal to extravasate, they are able to transform their morphology and flatten onto the endothelium surface. This transition, from a spherical to a flattened morphology, is the first key step which neutrophils undergo before moving out of the blood and into the extravascular tissue space. Once they have migrated through tissues towards sites of infection, neutrophils carry out their primary role—killing infecting microbes by performing phagocytosis and producing toxic reactive oxygen species within the microbe-containing phagosome. Phagocytosis involves the second key morphology change that neutrophils undergo, with the formation of pseudopodia which capture the microbe within an internal vesicle. Both the spherical to flattened stage and the phagocytic capture stage are rapid, each being completed within 100 s. Knowing how these rapid cell shape changes occur in neutrophils is thus fundamental to understanding neutrophil behaviour. This article will discuss advances in our current knowledge of this process, and also identify an important regulated molecular event which may represent an important target for anti-inflammatory therapy.

## 1. Introduction

When neutrophils undergo spreading on the endothelium (or experimentally, on other surfaces) or phagocytosis, there is an apparent expansion of the plasma membrane by nearly 200% [1,2]. However, because the plasma membrane is mainly composed of a phospholipid bilayer, interspersed with proteins and other lipids, it is not able to ‘stretch’ very much. The hydrophilic head groups of the phospholipids, which face the extracellular and intracellular water phases, tightly sandwich the hydrophobic fatty acid chains between them and provide a strong barrier between the two water interfaces. However, adjacent phospholipids interact weakly with each other, so there is little lateral strength. It is estimated that a phospholipid bilayer membrane can stretch laterally by only about 4% before rupturing [3]. Understanding the mechanism by which this rapid and large apparent expansion of the plasma membrane in neutrophils occurs is, therefore, a key question in understanding neutrophil spreading and phagocytic activity.

A number of mechanisms which could provide additional plasma membrane for this process are all unlikely to operate within neutrophils [1,2]. Fusion of the forming phagosome with juxta-plasma membrane endoplasmic reticulum (ER) membrane may provide ‘additional membrane’ in macrophages [4], but it is unlikely to account for the membrane required for cell spreading, which would result in the inversion of a major fraction of ER throughout the cell. In neutrophils, there is, in any case, very little ER [5] and none near the plasma membrane [6]. It is similarly unlikely that localised exocytosis at the forming phagocytic cup may provide the additional membrane. The surface area of each granule would provide less than 0.1% of the membrane required for cell spreading, and this would consume all of the granular membrane content [1,2]. Explaining the origin of the additional membrane may be simpler when the geometry of the neutrophil surface area is considered more realistically. Although light microscopy suggests that, in the circulation, neutrophils have an apparently spherical morphology. Scanning electron microscopy reveals that the surface of the ‘sphere’ is highly wrinkled and has many surface ‘microridged’ structures [5,7]. It has been estimated that the surface of a neutrophil contains approximately 85% more plasma membrane than is required to enclose the cell volume [8]. Upon osmotic swelling, and neutrophil spreading, these surface wrinkles in the membrane are reduced. The potential membrane stored in these membrane microridges could contribute 100% additional cell surface area [9], therefore these structures have sufficient membrane to act as the ‘membrane reservoir’ once they have flattened (Figure 1). Thus, the question of how neutrophils increase their apparent cell surface area is now reduced to understanding how the cell surface wrinkles or microridges are maintained, and how cell-signalling induces them to be released. 

## 2. Biophysics of Cell Surface “Wrinkles”

A number of elegant studies which characterise the biophysical properties of the neutrophil plasma membrane have been made. Essentially, these studies used micropipettes attached to a vacuum pump to draw the plasma membrane up into the micropipette tip. This enables both the tension in the cell membrane and the amount of membrane available to be measured. Moderate suction can draw some of the neutrophil membrane up into the micropipette, equivalent to 5% expansion of the neutrophil membrane [10,11,12]. However, with greater suction pressure, significantly more membrane can be drawn up [13]. These findings were consistent with the ‘molecular Velcro’ model, based on the analogy with “Velcro”, a temporary clothes fastener with a strip of tiny hooks linking with a strip of tiny loops to give temporarily a strong join until pulled apart with increased force. In the neutrophil, the ‘molecular Velcro’ is sufficiently strong to hold the microridges in place against physiological osmotic pressure and maintain tension in the plasma membrane. The biophysical measurements show that there is some slack in the cell membrane, but once the slack is removed, additional force is can “tear apart” the ‘molecular Velcro’ holding the microridge structures in place [2,13]. 

In non-muscle cells, such as neutrophils, actin exerts a pushing force on the plasma membrane, rather than the contractile force of actin-myosin in muscle cells. This force is achieved by polymerisation of monomeric actin pushing against the plasma membrane. In neutrophils, there is a significant amount of polymerised actin beneath the cell cortex [14,15], which provides anchorage for filamentous actin growing towards the plasma membrane (Figure 2). The growing tips of these actin filaments continue until they reach the plasma membrane (Figure 2). Additional actin monomers can then only be added as a result of Brownian motion-driven fluctuations, in the bending of the plasma membrane [16]. If a gap between the tip of the actin filament and the plasma membrane opens up sufficiently for an additional actin monomer to enter, monomeric actin will be added to the existing f-actin chain. The plasma membrane cannot then return to its original position, and, as a result, the membrane has been pushed out a little further (Figure 2). This mechanism is a Brownian ratchet [16] and its theoretical maximum rate of unimpeded pushing force against the plasma membrane is 0.75 μm/s, with an actin concentration of 25 μM [16]. This exceeds that required for the fast rate of neutrophil spreading, at about 10 μm/100 s (i.e., 0.1 μm/s). The process continues until the membrane tension exceeds than that which allows Brownian fluctuations in the position of the membrane (Figure 2). However, once the membrane tension is reduced (such as when the links which secure the membrane microridges are released), the Brownian ratchet mechanism will continue to push against the membrane, forming protrusions. This effect probably underlies an observation made on HL60 neutrophils, where cells were allowed to stretch to a point where the cell body and the motile and ruffling front of the cell were separated only by a thin tether of cell membrane [17], at which point the ruffling at the front of the cell ceased. In this case, the highly stretched cell morphology had presumably utilised all the membrane reservoir and the tension at the front of the cell was sufficient to prevent the Brownian ratchet and actin polymerisation. When the experimenters broke the tether by laser cutting, the front of the cell immediately began to ruffle, as expected upon reduction in membrane tension and the availability of additional membrane. Thus, an important function of the microridges is to both maintain a membrane reservoir and also limit cortical actin polymerisation. Once the microridges are allowed to unfold there will be both the additional membrane required for pseudopodia formation, or cell spreading, and the enabled actin polymerisation Brownian ratchet will push out the plasma membrane. An important finding came from measurements of the force required to deform the plasma membrane into a micropipette tip [11]. It was found that the ‘Velcro-like’ adhesion within microridges was considerably reduced during neutrophil phagocytosis [11]. This was consistent with the release of the ‘molecular Velcro’ during phagocytosis, suggesting that the Brownian ratchet would thus be able to continue to deform the plasma membrane. 

Scanning electron microscopy reveals that the plasma membrane which forms the phagocytic cup is devoid of microridges [5,18], in contrast to the highly wrinkled cell body. A similar conclusion can be drawn from living neutrophils interrogated by subdomain fluorescence recovery after photobleaching, or sdFRAP [19,20]. This optical technique measures the time taken by a membrane-associated fluorophore from the boundary of a zone of photobleaching to translocate to a subdomain distant from the bleach front. This gives information about the actual diffusion distance, and thus the flatness of the surface over which diffusion has occurred. In neutrophils, the diffusion distance of the cell body is significantly greater than that expected for a flat surface (but not unexpected for a highly wrinkled surface), whereas at the phagocytic cup, it is consistent with diffusion over a smooth membrane surface [20]. Thus, both in neutrophils fixed for SEM and in living cells, the cell surface topography is demonstrably altered during phagocytosis.

## 3. Signalling the Release of Cell Surface Microridges/Wrinkles

It has long been known that immediately before spreading onto a surface, there is a transient rise in global cytosolic free Ca^2+^ within the neutrophil or macrophage [21,22]. This Ca^2+^ signal is probably induced by immobilisation of cell surface adhesion molecules, such a β2-integrin (CD11b/CD18). Experimentally, immobilising these molecules alone causes a Ca^2+^ signal [23,24,25]. This Ca^2+^ signal is not a consequence of spreading, but rather the cause, because (i) it occurs before the cell spreads and (ii) a cytosolic Ca^2+^ signal alone can induce cell spreading. For example, without integrin engagement, experimentally-induced Ca^2+^ signals are able to cause neutrophil spreading in the absence of other stimuli. Neutrophil spreading occurs in response to experimentally-induced elevations of cytosolic Ca^2+^ achieved by either photolytic uncaging of cytosolic caged Ca^2+^ [26] or cytosolic caged IP_3_ [27,28]. This provides strong evidence for the direct role of an elevation in cytosolic free Ca^2+^ as the trigger for neutrophil spreading.

However, it has been more difficult identifying the Ca^2+^-activated enzyme responsible for this effect. The use of pharmacological inhibitors has eliminated a number of potential Ca^2+^ targets including calmodulin, protein kinase C (PKC) and calcineurin. However, a number of chemically unrelated inhibitors of the Ca^2+^ activated protease, µ-calpain, inhibit neutrophil spreading [27,28]. µ-Calpain is a member of a family of cysteine proteases which are expressed within the cytosol rather than within lysomes or other organelles. µ-Calpain is activated by cytosolic Ca^2+^ and is thus considered to be a regulatory protease rather than having a simple degradatory role. Its name reflects this, being a portmanteau word combining the molecular element for Ca^2+^ regulation (from calmodulin) and cysteine protease activity (from papain). Like many other proteases, µ-calpain is promiscuous, with no identified amino acid sequence required for recognition of the cleavage site [29]. However, in situ, the substrates of µ-calpain proteolysis are mostly cytoskeletally-associated proteins, such as vimentin, talin, merlin and spectrin [29]. There are now some highly specific novel inhibitors of the Ca^2+^ activation site on µ-calpain [30,31,32] which inhibit neutrophil spreading. Lymphocyte spreading on β_2_-integrin adhesion has also been shown to depend on µ-calpain activity [33,34]. By close observation of neutrophil pseudopod extension around a particle for phagocytosis, the timing of the Ca^2+^ signal is seen to proceed immediately before the acceleration phase of engulfment, [28,35,36] and the link between Ca^2+^ and acceleration is broken by inhibition of µ-calpain activity [28]. In µ-calpain null mice, neutrophils also fail to spread effectively and are aberrant in transendothelial migration [37]. Together, this evidence points towards the Ca^2+^ activation of µ-calpain as the signalling axis which leads to cell surface “unwrinkling”. 

In order to understand the mechanism by which activated µ-calpain mediates the release of cell surface microridges, and so permit cell spreading, we must first consider the molecules involved in maintaining the cell surface microridges.

## 4. Molecular Anatomy of Plasma Membrane Microridges

Cell surface wrinkles and microridges are maintained by proteins, especially of the ezrin/radixin/moesin (ERM) family [38]. These crosslink the cortical actin network to the plasma membrane and thus constitute the molecular identity of the components of the ‘molecular Velcro’ (Figure 3). This crosslinking connects the membrane to the underlying cortical actin in various non-planar configurations, including wrinkles and microridges (Figure 3). Such a cross-linking protein must have an actin-binding domain at one end of the molecule and a membrane binding domain at the other end. These are the characteristics of ERM family members of membrane-cytoskeletal linker proteins. Only two members of this family, have been identified in neutrophils: ezrin and moesin. Ezrin is abundant at the neutrophil periphery. It is responsible for producing and maintaining the intestinal epithelial villi [39,40], and may thus be an important protein in maintaining neutrophil cell surface microridges. It is, therefore, important to consider the structure and function of ezrin in detail. Within dynamic structures of the plasma membrane, such as in the microvilli of gastric parietal cells, the ezrin to actin ratio is almost 1:1 [41]. Given its relative abundance at sites proximal to dynamic cytoskeletal structures, it is likely that ezrin is involved in actin/plasma membrane re-organisation in neutrophils.

## 5. Ezrin 

Ezrin, previously known as p81, the epidermal growth factor (EGF) receptor tyrosine kinase substrate, villin and cytovillin is a member of the ERM family of membrane-cytoskeletal linker proteins. Ezrin was named in tribute to Ezra Cornell, a co-founder of Cornell University where it was first purified from the microvilli of epithelial cells [42]. Ezrin and moesin are the only members of the ERM family present in neutrophils. 

The phosphorylation status of ERM proteins correlates with their functional activity as crosslinkers between the plasma membrane and the actin cytoskeleton. Ezrin has two main functional domains, located at the N- and C- termini (Figure 4). When phosphorylated on either tyrosine or serine and threonine residues by protein kinase C, MRCK and AKT, ezrin is found associated with the cortical cytoskeleton [43,44]. In the non-phosphorylated state, ezrin is inactive and remains cytosol [45], where the N- and C-terminal domains are self-associated [46]. This finding led to the suggestion that in unbound monomeric ezrin, head-to-tail association of the N- and C-terminal domains inhibits the binding activity of either domain. In fact, as it is known that ezrin exists in two conformational states, conversion between the two is proposed to act as a switch to regulate ezrin activity [47]. The first state is an unphosphorylated closed, inactive conformation where the N-terminal ERM domain self-associates with the C-terminal actin binding domain [48], masking both plasma membrane and actin binding sites. The second is an open, active conformation where the two functional domains are separate. The open conformation is achieved through phosphatidylinositol 4,5-bisphosphate (PIP_2_) binding [40] and phosphorylation at the conserved threonine 567 (Thr567) residue, [44,49,50] which causes a change in the intramolecular binding properties between the two domains, facilitating ezrin self-dissociation and subsequent F-actin binding [51]. 

Ezrin was first suggested to play a role in crosslinking the plasma membrane to the F-actin cytoskeleton by Gould and colleagues [52] on account of its similarity to talin and band 4.1, which binds to glycophorin C [53]. Algrain and colleagues then performed transfection experiments which confirmed the interactions between the N-terminal FERM domain of ezrin and the plasma membrane [54]. They found that the N-terminal domain localised to the dorsal plasma membrane, and was readily extracted following treatment with non-ionic detergents, suggesting interaction with the plasma membrane [54]. As the ERM domain of ezrin forms complexes with the cytosolic tail of L-selectin [55], it also acts as an extracellular marker of bound cytosolic ezrin. L-selectin is found mainly on the crest of wrinkles and microridges of neutrophils and less abundantly in the ‘valleys’ between microridge [56], suggesting that ezrin too is mainly located within the microridges (Figure 3). The C-terminal actin-binding domain of ezrin binds to the F-actin cytoskeleton, which co-localises with actin filaments beneath the plasma membrane and remains associated with these even following treatment with detergent [54]. This affinity of ezrin for both the cortical actin cytoskeleton and the plasma membrane demonstrates a role of ezrin as a bridge between the two structures. Interestingly, mutations in the ezrin-binding domain of L-selectin impair neutrophil function rolling and adhesion, both in vitro and in vivo [57,58], although it is unknown whether this had any effect on the membrane reservoir, cell morphology or cell surface topography. An important additional effect of ezrin crosslinking is inhibition of the Brownian ratchet, which drives actin polymerisation (Figure 2 and Figure 4). By crosslinking to the underlying cortical actin network, tension in the plasma membrane is increased, and Brownian fluctuations are prevented (Figure 2 and Figure 4). The wrinkles and microridges are thus stable structures on the neutrophil surface, in which actin polymerisation is held in check so that the morphology of the microridge or wrinkle persists. Together, this points to a key role of ezrin, (or another linker protein between the plasma membrane and the cytoskeleton) as being important for maintaining the wrinkled cell shape and involved in the control of morphological cell shape changes.

ERM proteins bind to phosphatidylinositol 4-phosphate (PIP) and PIP_2_ in the plasma membrane [59,60], which also enables Thr567 phosphorylation [50]. Thus, Thr567 phosphorylation may stabilise the link between F-actin and the plasma membrane, stabilising the cell surface microstructures, such as microridges. Recently, it has been shown that an inhibitor of ezrin phosphorylation at Thr567, NSC668394, reduces the amount of peripherally located ezrin in neutrophils [61,62]. However, it has little effect on the rapid cell shape changes required for phagocytosis, suggesting that phosphorylation is not the regulatory step in neutrophils [61,62]. In contrast, elevation of cytosolic free Ca^2+^ and the activation of protease μ-calpain is important. 

## 6. µ-Calpain

µ-Calpain is a member of the calpain family of 15 known isoforms. The two most studied members are µ-calpain (also known as calpain-1) and m-calpain (also known as calpain-2), which share 62% sequence similarity [63]. µ-Calpain is a heterodimeric protein composed of a large 80 kDa subunit (CAPN1), encoded on chromosome 1, and a smaller 28 kDa regulatory subunit (CAPNS1) encoded on chromosome 19, in humans [29]. Domains V and VI of the small regulatory subunit, common to both μ- and m-calpain have a calmodulin-like region, consisting of five EF hand motifs which bind Ca^2+^ [64,65] Domain I of the larger subunit does not have any sequence homology to any other known protein. The proteolytic activity of µ-calpain arises from its catalytic triad of histidine, asparagine and cysteine in domain II of the larger subunit, which together attack a carbonyl group in the target substrate to hydrolyse it [63]. Domain II is also the region that interacts with the endogenous μ-calpain inhibitor, calpastatin. Domain III of the large subunit may have some regulatory function. It has a C2-like domain which enables µ-calpain to translocate to the plasma membrane through binding to phosphatidylserine in a Ca^2+^-dependent manner. This allows µ-calpain to associate with the plasma membrane in regions of elevated Ca^2+^ concentration [66]. Domain IV of the large subunit is partially homologous to the Ca^2+^ binding protein calmodulin, and contains four EF motifs, similar to domain VI of the smaller subunit.

µ-Calpain is a modulator protease rather than a degradative protease. Calpain modifies the activity or the behaviour of its substrate protein, rather than totally destroying it. However, experimentally, more than 100 substrates of µ-calpain have been reported. Although these can be broadly categorised into three main groups, including (i) transcription factors, (ii) kinases/phosphatases, and (iii) cytoskeletal and membrane-associated proteins [63]. The role of µ-calpain in neutrophil cell morphology change must clearly involve the last category, and especially a crosslinker protein of the ERM family. Neutrophils only express moesin and ezrin, of the ERM family of proteins. It is also significant that only ezrin, and not the structurally similar moesin, is susceptible to cleavage by µ-calpain [67]. Although the exact cleavage site is not known, analysis of the sequence differences between µ-calpain-sensitive ezrin and µ-calpain-insensitive moesin predict the cleavage to occur in the unique linker region of ezrin, between the two functional binding domains [60]. µ-Calpain cleavage of ezrin at this site breaks the F-actin-plasma membrane linkage and permit the Brownian ratchet to polymerise actin, forcing the plasma membrane away from the underlying actin cortex (Figure 5). As actin chains increase, the addition of WASP proteins may also occur, providing branch points for lateral actin polymers to form and so increase the outward pushing force (Figure 5). This mechanism exists in other cells such as endothelial cells, where ezrin cleavage by µ-calpain completely disrupts ezrin-actin interactions and results in membrane expansions called protrusions [68]. In gastric parietal cells, µ-calpain-mediated cleavage of ezrin causes reduced proton accumulation via microvilli [69]. In neutrophils, mild inhibition of µ-calpain slows down the spreading process such that it is possible to see the unfolding of the neutrophil membrane as bleb-like structures, presumably held by wrinkles which have yet to be released [27]. Activation of µ-calpain by an elevation of cytosolic Ca^2+^ has therefore been proposed as the mechanism for neutrophil membrane expansion and cell spreading.

## 7. µ-Calpain Activation by Ca^2+^ in Neutrophils

The suffixes of µ- and m-calpain indicate the difference in Ca^2+^ concentration required for in vitro activation, 20–60 µM Ca^2+^ for µ-calpain and 0.3–1.3 mM Ca^2+^ for m-calpain respectively [63]. Both Ca^2+^ concentrations exceed the global cytosolic free Ca^2+^ concentrations reported in neutrophils, which peaks at 1 µM (e.g., [70,71,72]). µ-Calpain translocates from the cytosol to the cell periphery via its C2-like domains, in response to elevated Ca^2+^ influx [73,74]. It is at the cell periphery that µ-calpain, if activated, would cleave ezrin. Thus, the Ca^2+^ concentration near the plasma membrane, rather than the concentration in the bulk cytosol, is important for activating µ-calpain for ezrin cleavage. Unlike the global Ca^2+^ concentration in the cell, Ca^2+^ near the influx channels can reach considerably higher concentrations. Using a fluorescent indicator trapped in the plasma membrane, [75] the Ca^2+^ concentration at the cell edge of neutrophils in suspension reaches high levels during Ca^2+^ influx, estimated to be greater than 30 µM [76]. This is not seen in neutrophils that are closely opposed to a surface, for total internal reflection fluorescence (TIRF) imaging [77], suggesting that the high peripheral Ca^2+^ concentration reported in neutrophils in suspension may be confined to the microridges (which are present on cells in suspension, but absent in cells which have spread or are tightly adherent). Mathematical modelling Ca^2+^ concentration has predicted that, if the Ca^2+^ influx channels were distributed uniformly over the neutrophil surface when the bulk cytosolic reached 1 µM, the concentration of Ca^2+^ that could be reached within the microridges would be very high [78]. Depending on the size and shape of the microridge, an upper estimate of 100 µM could be achieved [78]. Thus, intra-wrinkle concentrations of Ca^2+^ are predicted to be sufficient to activate µ-calpain locally at these Ca^2+^ hotspots, specifically, at locations were ezrin would also be expected (Figure 6). 

Although calpain activation has been demonstrated by immunoelectron microscopy in fixed cell [79], the measurement of µ-calpain activity within living neutrophils is difficult. However, cell-permeable fluorogenic µ-calpain substrates suggest that µ-calpain activity is elevated during cell spreading [27]. More specific, but non-permeant probes can be microinjected into neutrophils using a no-touch electroinjection technique [73,80], and show that µ-calpain activation can occur in living neutrophils, correlating with cell-spreading and phagocytosis [80,81]. As the fluorescent product of µ-calpain activity was free to diffuse through the cytosol, these studies could find no evidence of localised activation. However, with a large number of potential substrates in the cytosol, it may be considered important that µ-calpain is not activated by Ca^2+^ concentrations reached in the bulk cytosol and so activation *per se* must occur only at the cell periphery.

## 8. Conclusions

In this review, we have discussed the mechanism underlying the ability and triggering of the rapid cell shape changes which neutrophil undergo when spreading onto a surface or during phagocytosis. We have focused on (i) the signalling from an elevation of cytosolic Ca^2+^ to (ii) µ-calpain activation, leading to (iii) the cleavage of ezrin and (iv) the release of cell surface wrinkles and microridges which (iv) permit actin polymerisation at the cell cortex to push the unfolding plasma membrane and (v) result in a rapid transition into the spread morphology or the localised formation of pseudopodia required for phagocytosis (Figure 1). The result of releasing the ezrin–membrane link not only allows unfolding of cell surface wrinkles and microridges, but also allows the membrane between the microridges to be pushed out (Figure 7). This series of events thus gives a molecular explanation underlying the process of neutrophil cell shape change required for phagocytosis and cell spreading. It also points to a target for anti-inflammatory disease treatment, as inhibition of µ-calpain activation in neutrophils is expected to reduce the rate of neutrophil extravasation to inflammatory sites in the same way as anti-TNF therapy. Inflammation is reduced by inhibition of calpain in a number of experimental animal models of inflammation [82,83,84,85,86,87]. However, the pharmacological agents used lack sufficient specificity to be useable in humans. For this reason, there is considerable interest in designing µ-calpain inhibitors with high specificity [88]. A promising line of research is based on inhibition of the Ca^2+^ activation mechanism, rather than the protease domain, and a number of number compounds have been reported [29,30,31,89]. Hopefully, with a complete understanding of the cellular and molecular basis of inflammation and especially neutrophil trafficking, some inflammatory diseases may be preventable if treated at an early enough stage.

## Figures and Tables

**Figure 1 ijms-20-01383-f001:**
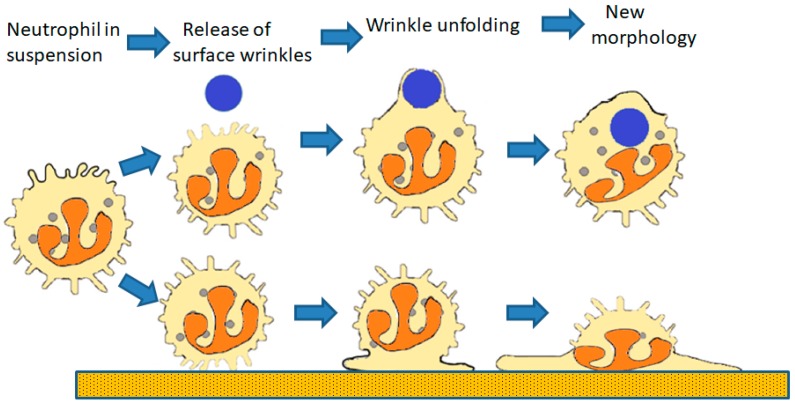
The role of the cell surface reservoir of membrane. The surface reservoir (wrinkles) permits cell spreading onto a surface (**lower sequence**) or for phagocytosis (**upper sequence**) in three steps. The first step (disconnecting surface wrinkles from underlying cortical actin) is the result of the release of the ‘molecular Velcro’ initially holding the wrinkles in place. The second step is the unfolding of the wrinkles as the result of Brownian ratchet driven actin polymerisation initiated by the slackening of membrane tension. The last step results in the final cell configuration with the additional membrane employed to form the phagosomal.

**Figure 2 ijms-20-01383-f002:**
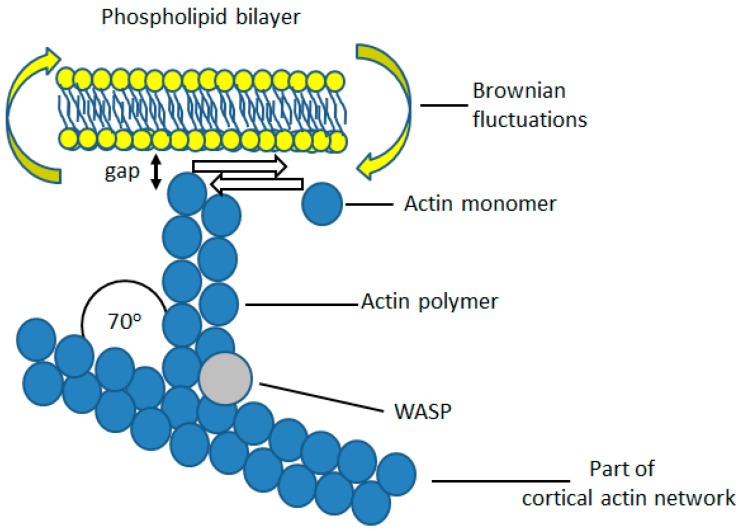
The Brownian ratchet for actin polymerisation near the plasma membrane. From the cortical actin network, branch points are formed by insertion of WASP protein which allows an additional point for another actin polymer to grow towards the plasma membrane. This actin polymer continues to grow until it encounters the plasma membrane and the gap between the polymer tip and the plasma membrane is less that the diameter of a single actin monomer. However, if there is slack in the membrane, Brownian fluctuations in the position of the plasma membrane will occur (as the membrane moves back and forth randomly) and the gap may transiently be greater than the diameter of an actin monomer. In which case, the actin polymer will grow and ‘push’ the membrane. The Brownian ratchet will continue until the tension in the plasma membrane increases to a point when Brownian fluctuations are so lessened that no gap which is greater than the actin monomer size can form between the actin polymer tip and the plasma membrane.

**Figure 3 ijms-20-01383-f003:**
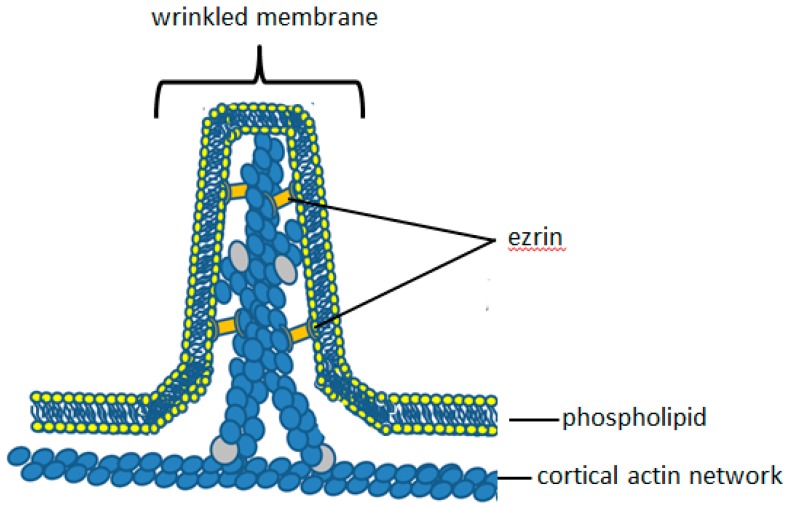
The molecular anatomy of the neutrophil cell surface wrinkles. The relative locations of ezrin (within the wrinkles) and the cortical actin network are shown. As in Figure 3, the ezrin crosslinks polymerised actin to the plasma membrane and prevents the operation of the Brownian ratchet, which drives actin polymerisation to push out the plasma membrane. The wrinkle is consequently a stable structure on the cell surface, which is little affected by Brownian effects.

**Figure 4 ijms-20-01383-f004:**
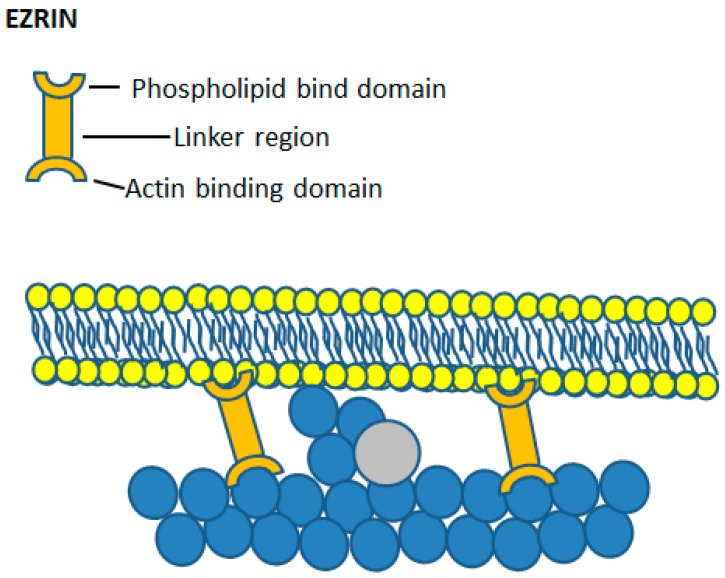
Ezrin crosslinking actin and the plasma membrane. The upper cartoon of an ezrin molecule shows the three main features—the phospholipid binding domain, the actin-binding domain and the linker region between the two. Beneath this is an illustration depicting the effect of ezrin crosslinking on the ability of actin to polymerise. With ezrin binding the cortical actin network to the plasma membrane, Brownian fluctuations in the plasma membrane are locally prevented and actin polymerisation cannot occur. In this way, ezrin binding stabilises cell surface structures such as villi and neutrophil wrinkles.

**Figure 5 ijms-20-01383-f005:**
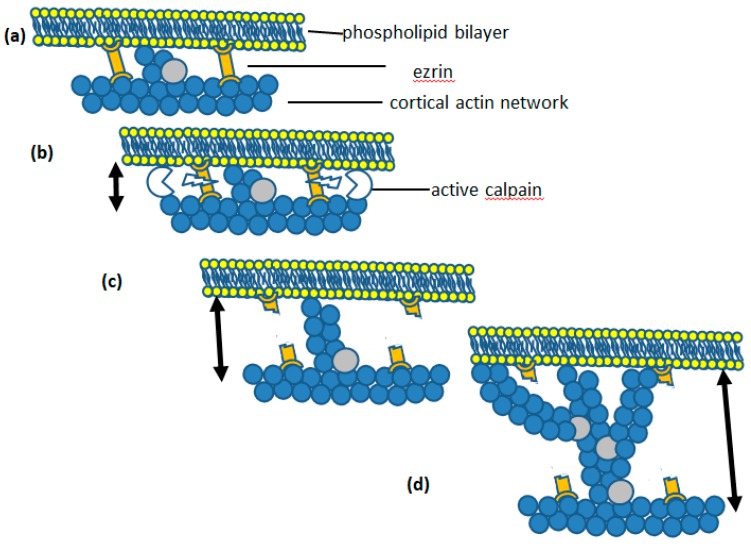
How ezrin cleavage initiates actin polymerisation. In (**a**), the position of the membrane in relation to the plasma membrane is stable and is also shown in Figure 2. In (**b**), the ezrin linkage is broken by the action of activated µ-calpain, which results in (**c**) the establishment of the Brownian ratchet and the growth of actin polymers ‘pushing out’ the plasma membrane. In (**d**), the actin polymer is sufficiently long enough for branch points to be added actin polymerisation continues to push out the plasma membrane. The arrows at the side of a, b, c and d show the distance from the plasma membrane to the cortical actin network increasing.

**Figure 6 ijms-20-01383-f006:**
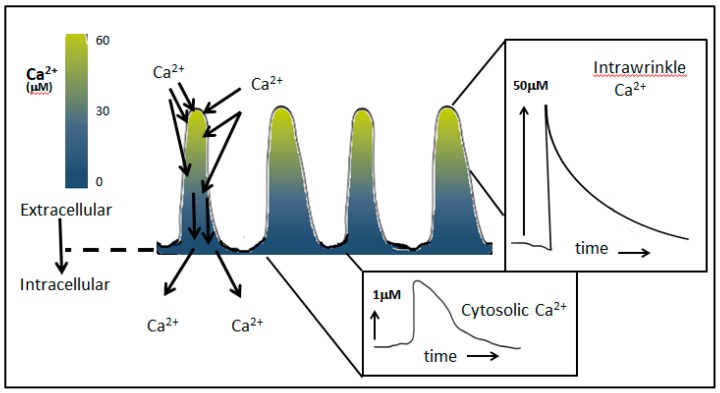
Intra-wrinkle Ca^2+^ reaches a concentration high enough to activate µ-calpain. Mathematical modelling [78] shows that within the folded region of the plasma membrane, Ca^2+^ concentrations reach very high levels, at least as high as that required for calpain activation. This is because the limited volume of the wrinkles has a relatively large surface area, so the effect of Ca^2+^ influx is exaggerated within the wrinkle as compared to the whole cell. Once local Ca^2+^ buffers within the cytosol of the wrinkles are saturated, the elevation of Ca^2+^ is limited only by the rate of diffusion of new mobile Ca^2+^ buffers into the wrinkle. The figure shows a pseudo-coloured representation of the Ca^2+^ concentration, together with examples of the expected Ca^2+^ transients within the wrinkle and within the cell.

**Figure 7 ijms-20-01383-f007:**
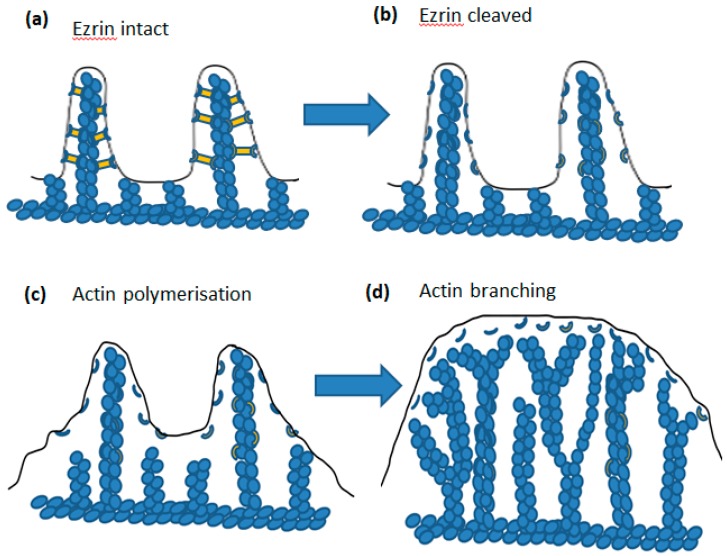
The sequence of intracellular molecular events leading to ”membrane expansion”. In (**a**) the wrinkles are held in place (more detail in Figure 4). Following cleavage of ezin by activated µ-calpain (as shown in Figure 4), the tension in the membrane is relaxed and (**c**) shows that result on actin polymerisation, which can now proceed via the Brownian ratchet mechanism. In (**d**), actin polymerisation has progressed and branch points added such that the membrane made available by wrinkle detachment is pushed out. During phagocytosis, this would be coordinated to form a phagocytic cup, as a result of localised adhesion via integrin or antibody on the particle and during cell spreading coordinated to spread onto the contacting substrate.

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
