# Peer review of "Neutrophil Cell Shape Change: Mechanism and Signalling during Cell Spreading and Phagocytosis"

_ijms, 2019, doi:10.3390/ijms20061383_

Round 1

Reviewer 1 Report

This review describes the mechanism underlying rapid cell shape changes of neutrophils, focusing on the role of calpain-mediated breakage of ezrin. It is quite interesting review, and the reviewer enjoyed reading it.  To improve the article, the reviewer has several comments as follows.

(1) If the authors would add information about “Velcro”, then the readers may understand easily what is suggested by the word “molecular Velcro”.

(2) In line 213, “glycophrin C” should be “glycophorin C”.  

(3) Perhaps the readers who are not familiar with calpain may need more basic information regarding implication of the name “calpain” and substrate “specificity”, for instance.

(4) The article of Saido et al. in JBC 268, 7422, 1993 may be useful to detect activated mu-calpain in the cells.

(5) There are some typographical errors and incomplete sentences, such as those in line 8, Figure 1 legend, line 88, line 140, line 198 and line 320.

(6) The authors may have to check the rule of the journal on where the reference number should be placed.  The journal may ask you to place in front of period but not after period.  

Author Response

We are very grateful to the reviewer for this time spent on this and for making comments which are aimed at improving our review.

(1) If the authors would add information about “Velcro”, then the readers may understand easily what is suggested by the word “molecular Velcro”.

 “Velcro” is a fastener consisting of a strip with tiny hooks that link temporarily to a strip with smaller loops until pulled apart. We have now added a simply explanation of the “molecular velcro” concept, namely that the cross-linking role of interaction ezrin is based on electrostatic (rather than covalent) binding which, though strong,  can be  broken by physical force. Velcro provides an analogy of this.

(2) In line 213, “glycophrin C” should be “glycophorin C”.  

This has now been corrected (on new line  216) 

(3) Perhaps the readers who are not familiar with calpain may need more basic information regarding implication of the name “calpain” and substrate “specificity”, for instance.

An explanation of the name and its specificity of calpain have now been given on lines 158-162.

(4) The article of Saido et al. in JBC 268, 7422, 1993 may be useful to detect activated mu-calpain in the cells.

 Thank-you for the suggestion. We have now added this to the review.

(5) There are some typographical errors and incomplete sentences, such as those in line 8, Figure 1 legend, line 88, line 140, line 198 and line 320.

 All typos that we have spotted (including those specifically listed) have now been corrected.

(6) The authors may have to check the rule of the journal on where the reference number should be placed.  The journal may ask you to place in front of period but not after period.  

We have re-positioned reference numbers so that they are now before a final period or other punctuation marks.

Reviewer 2 Report

This review is a comprehensive summary of the molecular mechanisms at the basis of the rapid cell shape changes which neutrophil undergo while exerting their functions, like spreading and phagocytosis. The interesting aspect is the possibility that proteins involved in these processes may be the target of therapeutic intervention in inflammatory situations in which neutrophils takes large part. I suggest to add a small paragraph about this last point to underlie the importance of the knowledge of this sophisticated mechanisms. Some typos are present.

Author Response

I suggest to add a small paragraph about this last point (therapeutic intervention) to underlie the importance of the knowledge of this sophisticated mechanisms.

We have expanded a little on this point (in lines 368-373 , with 6 extra refs), which we agree is important (but not the subject of this review)..

Some typos are present.

All typos that we have spotted have now been corrected.